# Evolution of the ionisation energy with the stepwise growth of chiral clusters of [4]helicene

Sérgio R. Domingos [1,7] ✉, Denis S. Tikhonov [1] ✉, Amanda L. Steber [1,8] ✉, Patrick Eschenbach[2,3], Sebastien Gruet[1], Helgi R. Hrodmarsson [4,9], Kévin Martin[5], Gustavo A. Garcia [4], Laurent Nahon [4], Johannes Neugebauer [2,3], Narcis Avarvari [5] & Melanie Schnell [1,6] ✉

Polycyclic aromatic hydrocarbons (PAHs) are widely established as ubiquitous in the interstellar medium (ISM), but considering their prevalence in harsh vacuum environments, the role of ionisation in the formation of PAH clusters is poorly understood, particularly if a chirality-dependent aggregation route is considered. Here we report on photoelectron spectroscopy experiments on [4] helicene clusters performed with a vacuum ultraviolet synchrotron beamline. Aggregates (up to the heptamer) of [4]helicene, the smallest PAH with helical chirality, were produced and investigated with a combined experimental and theoretical approach using several state-of-the-art quantum-chemical methodologies. The ionisation onsets are extracted for each cluster size from the mass-selected photoelectron spectra and compared with calculations of vertical ionisation energies. We explore the complex aggregation topologies emerging from the multitude of isomers formed through clustering of *P* and *M*, the two enantiomers of [4]helicene. The very satisfactory benchmarking between experimental ionisation onsets *vs.* predicted ionisation energies allows the identification of theoretically predicted potential aggregation motifs and corresponding energetic ordering of chiral clusters. Our structural models suggest that a homochiral aggregation route is energetically favoured over heterochiral arrangements with increasing cluster size, hinting at potential symmetry breaking in PAH cluster formation at the scale of small grains.

Helicenes were first discovered in 1903[1], resulting from *ortho*-condensation of aromatic rings: when at least four consecutive rings reach non-planarity to prevent steric hindrance, chiral helices are formed[2]. Axial chirality emerges from these helical topologies, with right- and left-handed helicenes labelled *P* and *M*, respectively (Fig. 1). [4]helicene is thus the smallest chiral carbon-based polycyclic aromatic hydrocarbon (PAH). Besides their applications to generate multifunctional molecules with tailored chiroptical properties[3,4], pure carbon-based

[1]Deutsches Elektronen-Synchrotron DESY, Notkestr. 85, 22607 Hamburg, Germany. [2]Organisch-Chemisches Institut, University of Münster, 48149 Münster, Germany. [3]Center for Multiscale Theory and Computation (CMTC), University of Münster, 48149 Münster, Germany. [4]Synchrotron SOLEIL, L'Orme des Merisiers, 91192 Gif sur Yvette, Cedex, France. [5]Univ Angers, CNRS, MOLTECH-Anjou, SFR MATRIX, 49000 Angers, France. [6]Institut für Physikalische Chemie, Christian-Albrechts-Universität zu Kiel, Max-Eyth-Str. 1, 24118 Kiel, Germany. [7]Present address: CFisUC, Department of Physics, University of Coimbra, 3004-516 Coimbra, Portugal. [8]Present address: Department of Physical Chemistry, Faculty of Science, University of Valladolid, 47011 Valladolid, Spain. [9]Present address: LISA UMR 7583 Université Paris-Est Créteil and Université de Paris, Institut Pierre et Simon Laplace, 61 Avenue du Général de Gaulle, 94010 Créteil, France. ✉e-mail: sergio.domingos@uc.pt; denis.tikhonov@desy.de; amanda.steber@uva.es; melanie.schnell@desy.de

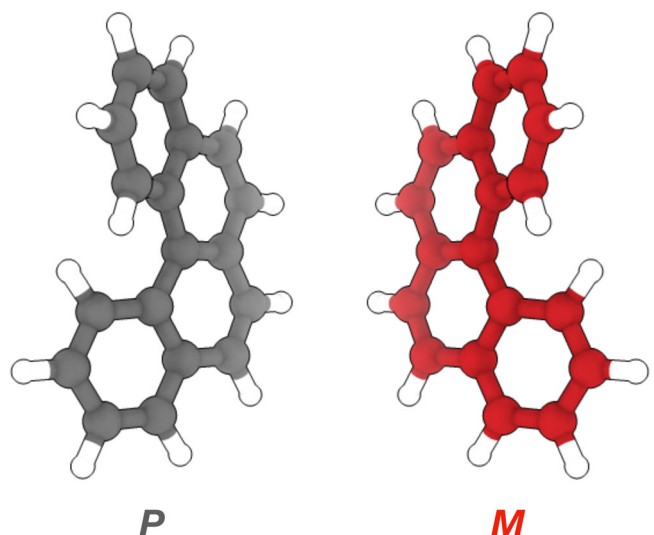

**Fig. 1 | Molecular structure of [4]helicene.** *P* and *M* enantiomers of the smallest PAH with helical chirality ($C_{18}H_{12}$).

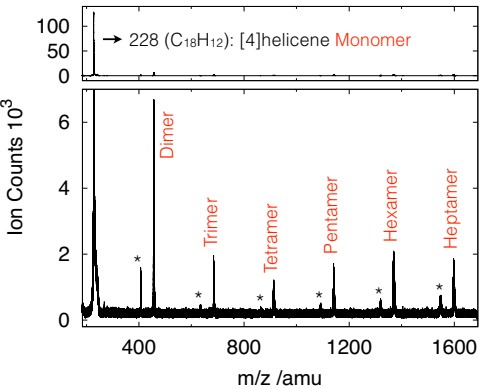

**Fig. 2 | Mass spectrum.** Depiction of the [4]helicene cluster distribution from the time-of-flight mass spectrometer obtained at 11 eV photon energy. The lower panel provides a vertical zoom. The mass peaks marked with an asterisk come from the residual vapour pressure of phenanthrene in the vacuum chamber, leading to the formation of [4]helicene–phenanthrene clusters.

helicenes have recently been added to the library of potential emitters of diffuse interstellar bands (DIBs)[5].

The exact size distribution and isomer constitution of astronomical PAHs are unknown. Yet, neutral or ionised PAH clusters are first-row candidates for molecular cloud grains[6]. PAH cations bind strongly to neutral PAHs by charge-induced bonding mechanisms, with longer interaction ranges than typical dispersive van der Waals interactions, having thus the potential to enhance PAH clustering and accelerate the formation of very small grains (VSGs)[7]. Laboratory data on these systems is thus in high demand, bridging the gap between processes of PAHs and bulk-derived properties of classical grains. This information, expanding on the ionisation energies of PAHs in the framework of cluster formation, is key to construct plausible aggregation motifs and furthering our understanding of PAH space chemistry[7,8].

PAHs span a large family of systems, from the smallest molecules indene $C_9H_8$ and naphthalene $C_{10}H_8$ up to $C_{348}H_{48}$ (the largest known to date[9]). Searches of PAHs in the interstellar medium (ISM) take into consideration that spectral features may likely arise from smaller PAHs (with 50 carbon atoms or less) because their lower heat capacity will allow mid-IR emission[10]. Unambiguous assignment of PAHs from IR emission spectra is generally difficult, and with the exception of indene[11], recently detected in the TMC-1 cold dark molecular cloud via rotational spectroscopy and radio astronomy, no other pure carbon-based PAH has been identified in space. Of note are still two fullerenes: the Buckminsterfullerene $C_{60}$[12] and $C_{70}$[13], both detected in planetary nebulas. Very recently, three nitrogen-containing PAHs (1-cyano-naphthalene, 2-cyanonaphthalene, and 2-cyanoindene) were identified from their radio signatures using stacking and matched filter analysis[14,15], and these are inspiring future searches to detect other systems of this class. To support these astronomical searches, several laboratory rotational spectroscopy studies provide the data to assist in the assignment of new species in the ISM[16–19].

PAHs, and hence helicenes, are strong UV-absorbers thus causing the heating of gas in the ISM, especially via their photoemission[20]. It has been suggested that through analysis of the relative intensities of specific mid-IR spectral features, it becomes feasible to trace the ionisation state of PAHs and thereby explore their radiative environment[9]. In this framework, the size and clustering of PAHs is expected to complicate detection in the ISM since larger systems are cooler, and the emission magnitudes of aromatic-based IR bands are sensitive to temperature. On the one hand, clusters will naturally prompt weaker emissions than individual molecules. Theoretical studies using molecular dynamics simulations show that the aftermath of two colliding PAH molecules is likely a weakly bound cluster if the collision energy is low. In this scope, Rapacioli et al. proposed that to achieve a 50% probability of bouncing in coronene monomer collisions, the ambient gas requires a temperature of 13,500 K, with even higher temperatures necessary when larger structures are involved[21]. Moreover, clusters may dissociate under incident UV radiation, freeing individual molecules. On the other hand, ionised clusters are increasingly more attractive due to their higher stability, larger ionisation cross sections and lower ionisation energies than those of monomers[22].

While it is established that PAH clusters are important precursors of carbonaceous dust particles in the ISM[23], their formation mechanisms and interactions with VUV radiation are not well understood. Thus, it is of great importance for astrochemical models to elucidate the structure and dynamics of PAH clusters in the gas phase[24]. Here, results of a combined experimental and theoretical study on the ionisation energies of chiral clusters of [4]helicene up to the heptamer are presented. Through the introduction of handedness in cluster formation, this investigation extends our understanding of symmetry breaking and chiral amplification of large systems with rich PAH content in harsh vacuum environments.

## Results

### Photoelectron photoion coincidence spectroscopy

In Fig. 2 we show the mass spectrum depicting ion signals for the monomer and the complete cluster series up to the heptamer of [4] helicene. The expected decrease in ion signal with increasing cluster size reaches a turning point at the tetramer, where an increasing abundance for the following larger aggregates is registered. This behaviour may have several origins, but it is out of the scope of this paper. The photoelectron spectrum of the [4]helicene monomer following VUV excitation at 11 eV is shown in Fig. 3 (red trace), revealing the energy distribution of ejected electrons and disclosing the distribution of accessible energy levels of the ionised system (see also Supplementary Fig. 6). Overlaid (in black) are depicted the vibrationally resolved photoelectron spectra (PES) of the ground state plus the lowest five excited states of the [4]helicene monomer computed using time-dependent density functional theory (TD-DFT) at the PBE0[25]/def2-TZVPP[26] level of theory. A detailed account of the computational details is presented in the supplementary information file. Overall, the calculated spectrum captures the essential features observed in the experiment, with the energies of each of the electronically excited states being well predicted and their relative spacing captured. Several

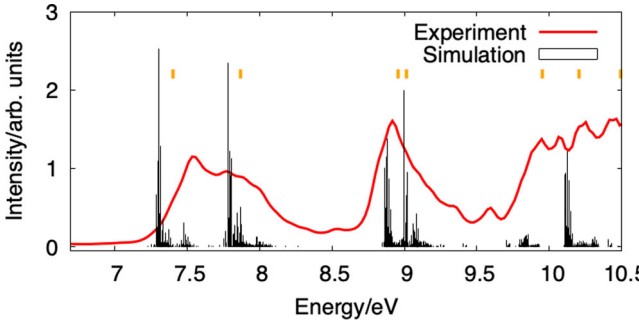

**Fig. 3 | Photoelectron spectrum of the [4]helicene monomer.** Vibrationally resolved photoelectron spectrum (PES) recorded after VUV excitation at 11 eV is shown (red trace). The calculations (black traces) were performed at the PBE0[25]/def2-TZVPP[26] level of theory. The neutral geometries of [4]helicene and its cation in the ground and first five excited states were optimised, and the Hessians at each equilibrium structure were computed. The resulting spectrum was computed using the ezFCF programme[49]. Vertical ionisation potentials (yellow markers) were calculated at the geometry of the neutral species.

vibrational progressions are visible, in particular for the higher-energy states, but their mode assignment is beyond the scope of this report.

The mass-channel selection allows one to isolate the detected ions and corresponding photoelectron spectra for each cluster size and with that, to evaluate the ionisation energies for each aggregate. In Fig. 4 we show the mass-selected photoelectron spectra produced via VUV excitation at 11 eV, recorded for approximately two hours. The experimental values for the ionisation onsets were retrieved using the following methodology: the low-energy edge of the PEPICO spectrum for each cluster size was fit with a function $I(\nu) = A + B \cdot (1 + \text{erf}(\frac{\nu - \nu_0}{\delta_\nu}))$, where $\nu$ is the photon energy, $A$ is the baseline intensity, $B$ is the increased height, erf(x) is the error function, $\nu_0$ is the rising edge inflection point, and $\delta_\nu$ is the increase width parameter. The ionisation thresholds were determined as the intersection of the tangent line at the inflection point with the baseline (see Supplementary Fig. 5). A detailed account of this determination is given in the supplementary information file.

Although mass selection does not necessarily fully translate to size selection due to dissociative photoionisation processes (DPI), we can safely assume that this will be the case close to the ionisation onsets. Indeed, in another π-stack system, namely pyrene[27], dissociative energies (difference between IE and fragment appearance energy) of more than 1 eV have been observed. Here, from the energies given in Supplementary Tables 2 and 6, one can extract the calculated dissociation energies for the process $M_n^+ \rightarrow M_{n-1}^+ + M$ for the most stable cluster at 1.04 eV for $n = 2$ and 0.81 eV for $n = 7$ for instance (see Supplementary Table 7). These values are most likely lower limits due to the short analysis time of the spectrometer (few μs), which translates into a so-called kinetic shift of the DPI process towards higher ionisation energies and to the possible existence of reverse barriers. Therefore, we believe that our onset values are not affected by DPI, since each cluster should be stable at least over the few 100 meV above its IE. Note that some of the oscillations visible in the clusters' spectra might be due to noise induced by the Abel inversion of the velocity map imaging (VMI) images, but also to possible vibrational progressions, to the presence of various conformation families or even to closely lying electronic states whose analysis is out of the scope of the present work. Overall, the observed ionisation energies show a relatively steady decrease with increasing cluster size, spanning 7.263 ± 0.007 eV for the monomer, down to 6.798 ± 0.016 eV for the heptamer.

## Theoretical modelling of [4]helicene clusters

The theoretical modelling of the aggregation topologies of [4]helicene presents a challenge in comparison to previous investigations of

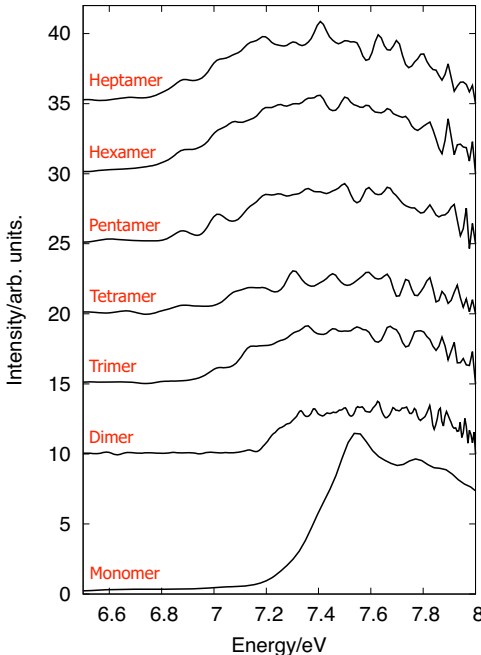

**Fig. 4 | Mass-selected photoelectron spectra (PES) at 11 eV photon energy.** The ionisation onsets for the [4]helicene monomer, and clusters up to the heptamer, are extracted in electron volt (eV): $E_1 = 7.26$, $E_2 = 7.16$, $E_3 = 6.94$, $E_4 = 6.86$, $E_5 = 6.88$, $E_6 = 6.78$ and $E_7 = 6.79$. Spectra are offset for clarity without any vertical scaling factor applied.

planar, and therefore achiral, PAHs[22]. Indeed, these clusters can assemble using combinations of $P$ and $M$ stereoisomers, leading to either homochiral aggregates, or a series of heterochiral clusters in different ratios of $P/M$ enantiomers, that can be characterised (similar to the enantiomeric excess quantity) with a degree of homochirality $\xi = \frac{n(P) - n(M)}{n(P) + n(M)}$, where $n$ denotes the total number of given enantiomer molecules in the cluster. The fact that several pairing options are available makes this study depart from a typical conformational-search problem, into a complex mapping of molecular cluster configurations.

In this work, we obtained equilibrium structures using two different computational strategies. Global geometry optimisations for the $n$-mers ($n = 1 - 7$) were performed using initial guesses predicted from both the ABCluster programme[28] and the Coalescence-Kick software[29]. The resulting structures were then optimised at the GFN2-xTB level of theory[30] and sorted by energy. For each cluster type, structures with relative energies above 300 K (210 cm$^{-1}$) were discarded from further analysis. Next, an algorithm was applied to all isomers of each cluster size to search for structurally similar clusters. When such a condition was met, the lowest-energy form was kept and the remaining ones were discarded. The computational details of the approach employed here are given in the supplementary information. The Cartesian coordinates for all relevant structures are given in the supplementary data 1. We note that conformers of the same cluster size can be close in energy, and consequently, using another method than xTB to sort the species could change which cluster is the lowest-energy form. We explore this in the supplementary information, where we employ an alternative energy ordering, according to the DLPNO-CCSD(T) level of theory (see Supplementary Table 6).

The results are condensed in Fig. 5, where relative energies and the degree of homochirality are depicted for each cluster size. Here, only the results for the paired enantiomers with $n(P) > n(M)$ are shown, as they would be the same as for their chiral counterpart with $n(M) > n(P)$. For example, the *PPM* trimer produces the same PES spectrum as the *MMP* trimer. The plot reveals a clear tendency for clusters to stabilise preferably in homoconfigurational motifs. With the exception of

the heptamer, the lowest-energy structure for all cluster sizes is homochiral. We note that the clusters with a degree of homochirality close to zero (e.g., racemic clusters) are typically predicted as the highest energy forms. Three-dimensional models of the lowest-energy clusters are shown in Figs. 6 and 7.

The dimer and trimer structures are shown in Fig. 6, in both homochiral and heterochiral configurations. *PP* depicts a stacked

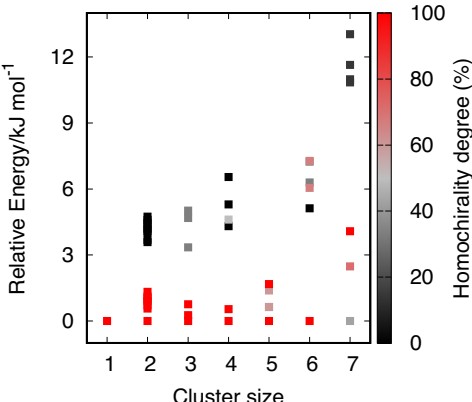

**Fig. 5 | Degree of homochirality as a function of [4]helicene cluster size.** Calculated electronic energies (GFN2-xTB) for all observed cluster sizes (data given in Table 2 of the supplementary information). The degree of homochirality $\xi = 100\% \times (n(P) - n(M))/(n(P) + n(M))$ is depicted for each cluster type, covering homochiral clusters (in red) and all possible combinations of heterochiral clusters (in grey) up to a racemic configuration (in black, e.g., *PPMM* for the tetramer).

arrangement with the same orientation for both monomers. In the *PM* dimer, opposite handedness prevents optimal stacking, and consequently, the "top" molecule rotates about 90° with respect to the "bottom" one. We note that similar stacking patterns were observed in previous studies with pyrene and coronene[22] as well as diphenyl ether, dibenzofuran, and fluorene[31]. The most stable homochiral trimer *PPP* shows a vertical stacking of subunits, while the heterochiral equivalent shows an intertwined *PM* and *PP* combination (see Fig. 6).

In Fig. 7, calculated structures for the homochiral tetramer, pentamer, and hexamer are displayed, which are the most stable isomers for each cluster size. The most stable isomer of the tetramer comes together as two consecutive, similarly oriented *PP* dimers, showing a vertically stacked topology. The same holds for the pentamer. However, here the orientation of subunits is not maintained overall: a three-over-two *PPP-PP* arrangement is in place, with ~180° rotation of the units that connect the two motifs (see Fig. 7). The homochiral hexamer, again the lowest-energy isomer of its size, organises in a tetramer-type motif plus a *PP* dimer that docks not on top but on the side of the stacked tetramer via CH − π interactions. Finally, the largest cluster detected experimentally in this study, the heptamer, is predicted to reach its most stable aggregate as a heterochiral cluster with *PPMPMPP* subunits arranged as a tetramer-type *PPMP* motif with a *MPP* trimer docked on the side via CH − π interactions (Fig. 7), breaking the trend of homochirality.

We used these equilibrium cluster geometries to calculate their respective vertical ionisation energies and assist in the interpretation of our PEPICO experiments. Figure 8 depicts the data set of ionisation onsets extracted from the mass-selected photoelectron spectra and the results from a series of calculations at different levels of theory. We

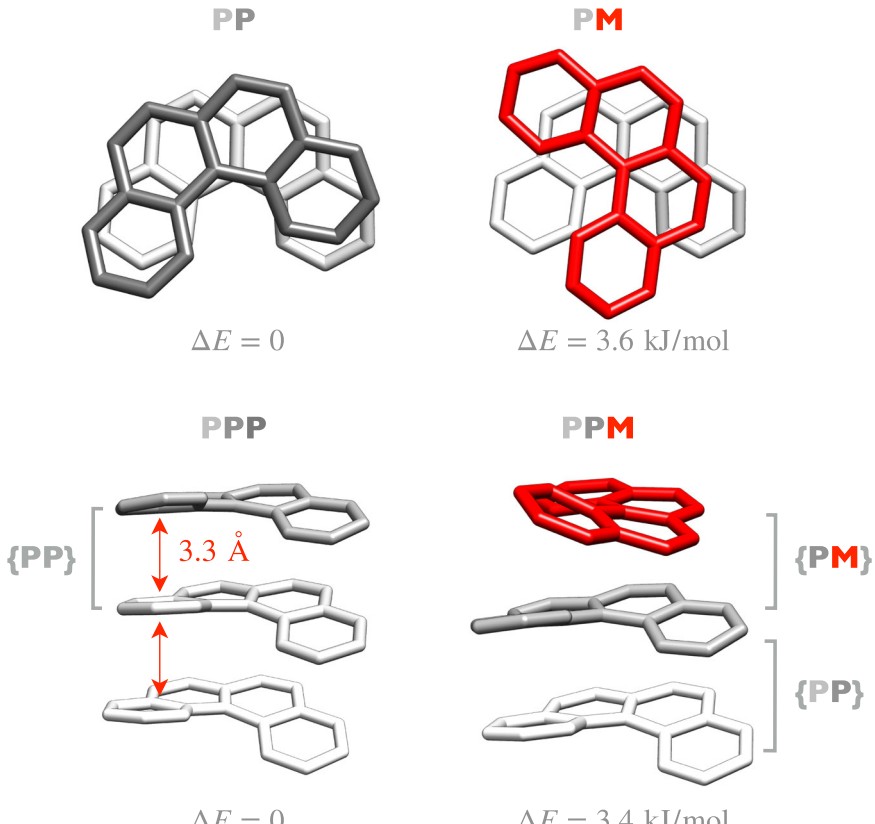

**Fig. 6 | Dimers and trimers of [4]helicene ions.** Calculated molecular structures of the most stable dimer and trimer topologies of [4]helicene ions for both homo-configurational (*PP* and *PPP*, respectively) and hetero-configurational (*PM* and *PPM*, respectively) aggregation. *P* is shown in grey, *M* is shown in red. Different shades of grey are used to facilitate visualisation of monomeric elements within the larger clusters. Structures were optimised at GFN2-xTB level of theory using the xTB software. Relative electronic energies are shown for each species. The brackets depict substructural pairs that match the dimer motifs.

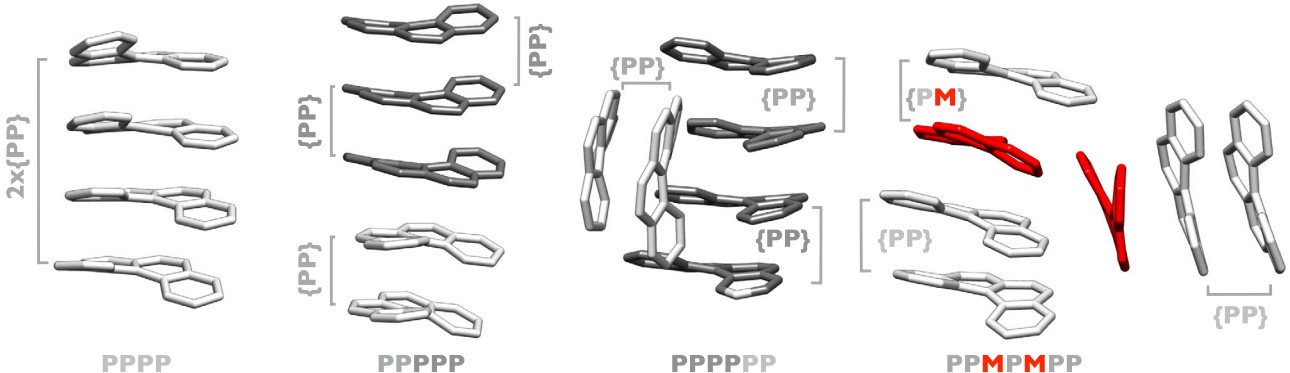

**Fig. 7 | Higher order (4≤N≤7) clusters of [4]helicene ions.** Calculated molecular structures of the most stable ionic clusters (tetramer, pentamer, hexamer and heptamer) of [4]helicene. Structures were optimised at the GFN2-xTB level of theory using the xTB software. The brackets depict substructural pairs that match either the homochiral (*PP*) or heterochiral (*PM*) dimer motifs. Different shades of grey are used to facilitate visualisation of monomeric elements within the larger clusters.

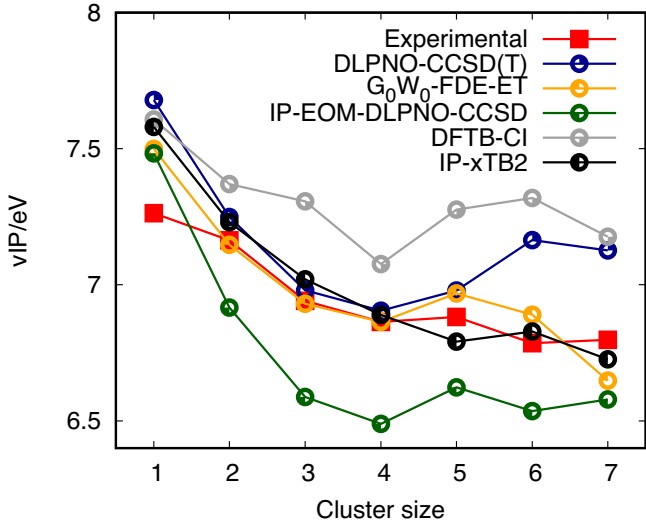

**Fig. 8 | Ionisation energies of [4]helicene clusters.** Evolution of the ionisation energies with cluster size of [4]helicene clusters (1 = monomer → 7 = heptamer). The data extracted from the experiment are compared with vertical ionisation potentials (vIP) for the lowest-energy isomers obtained at several levels of theory.

note that a shift between experimental and theoretical values is expected due to temperature effects (hot bands, presence of isomers) and potential differences between vertical and adiabatic ionisation energies. However, we find that regardless of the intrinsic limitations when comparing ionisation thresholds—obtained from the rising onset —to calculated vertical ionisation energies, the observed ionisation thresholds yield a trend that is consistently captured in the calculated vertical ionisation energies.

Vertical ionisation potentials (vIP) on the GFN2-xTB-optimised cluster geometries were calculated with high-level quantum-chemical methods (DLPNO-CCSD(T)[32–34], IP-EOM-DLPNO-CCSD[35–37], and a combination of frozen density embedding–electron transfer (FDE-ET)[38–41] and subsystem-based $G_0W_0$@BHLYP[42]) and with the computationally less expensive tight-binding-based semi-empirical methods IP-xTB2 and DFTB-CI[43]. The latter approach is comparable to the one employed in the previous study of planar PAHs[22], for comparison. The cc-pVTZ basis set was used for all calculations reported in this work. An in-depth analysis of the theoretical values is provided in the supplementary information (see Supplementary Figs. 2–4 and Supplementary Tables 1–5).

The size effect on the ionisation energy is captured by all the theoretical approaches we tested (Fig. 8). The apparent consistency found for all levels of theory employed likely emerges due to the GFN2-xTB-optimised geometries used for all calculations. The semi-empirical IP-xTB2 level of theory shows remarkable agreement with the experimental ionisation thresholds, but overall, all methods employed predict the experimental onset within ± 0.5 eV. The DLPNO-CCSD(T) method reproduces the experiment with accuracy for the dimer, trimer and tetramer, overshooting the observation for the larger clusters. The $G_0W_0$ − FDE-ET approach, combining two methods, has been tailored for the first time to describe a system of this class. In contrast to correlated ab initio methods, it opens up a route for reliable calculations for much larger clusters and at an acceptable computational timescale. The details of this approach are laid out in the supplementary information.

## Discussion

In summary, we present here an experimental PEPICO spectroscopy data set for chiral PAH clusters. The trend observed for the ionisation onsets with increasing cluster size is clear and in line with the single previous observation of such behaviour in PAH aggregates[22]. Here, however, computation and benchmarking of state-of-the-art theoretical methods quickly becomes a complex task. By introducing a combination of methods to explore conformational searches of large clusters, we provide a comprehensive comparison of theoretical methods to describe the decrease in the ionisation onsets with increasing cluster size. Quantitative agreement with experiment is challenging due to theoretical and experimental limitations linked to the size and complexity of the system. For instance, only experimental ionisation onsets can be extracted from the curves, which are subject to the presence of vibrational hot bands and conformer/isomer populations and thus are expected to differ from the calculated vIPs. Nevertheless, the combination of diabatic ionisation potentials obtained with subsystem-based $G_0W_0$[42] and electronic couplings from FDE-ET[38–41] led to a reliable description of the trend observed for the ionisation energies, introducing a novel methodology to tackle future investigations of increasingly larger and more complex systems.

On a concluding note, the theoretical methods employed consistently support our experimental determination of ionisation energies of [4]helicene clusters. At the same time, the agreement between experiment and theory also validates the structural models we developed including a complex grid of enantiomeric combinations of *P* and *M* configurations. These structural models, and their underlying building processes, pave the way for follow-up studies that may help accurately describing photo-ionised states of small grains as constituents in molecular clouds. Moreover, the methods employed disclose a clear preference for these clusters to stabilise

preferably in their homochiral configurations. Even though we cannot at this time provide further experimental validation for a prevailing homochiral aggregation route with the stepwise growth of helicene clusters, the results presented here suggest that work in this direction can potentially bring further insights on chiral preferences[44] and nucleation reactions[45] taking place in interstellar molecular clouds.

## Methods

### Experimental details

All photoelectron photoion coincidence (PEPICO) spectroscopy data reported here were acquired in a single measurement campaign using the DESIRS[46] vacuum ultraviolet (VUV) beamline at Synchrotron SOLEIL, where the DELICIOUS III double imaging PEPICO instrument[47] operates at the SAPHIRS permanent molecular beam end-station[48]. A sample of racemic [4]helicene was placed in a home-built stainless steel oven (in-vacuum) and heated to approximately 200 °C to generate sufficient vapour pressure. Here, the molecules have time to accumulate thermal energy in the vibrational modes to overcome the activation barrier of around 18 kJ mol$^{-1}$ (see Supplementary Fig. 1) following intramolecular vibrational redistribution, so that an equilibrium between $P$ and $M$ enantiomers is expected (see the supplementary computational details). The molecular gas was mixed with a constant flow of argon at 1.1 bar of stagnation pressure and adiabatically expanded[27] into the vacuum chamber through a 70 µm nozzle. At this stage, the molecular beam reaches temperatures below 100 K and the [4]helicene enantiomers will be energetically trapped, with considerably lower racemization process rates. Two consecutive skimmers with a 2 mm orifice are placed in front of the molecular jet expansion prior to entry into the ionisation zone of the spectrometer, where the molecular beam crosses the VUV synchrotron light. The DELICIOUS III consists of a velocity map imaging (VMI) instrument coupled in coincidence to a Wiley-McLaren time-of-flight (WM-TOF) 3D momentum imaging mass spectrometer. This configuration delivers mass-filtered photoelectron and photo-ion images simultaneously.

### Computational details

Structure optimisation and harmonic frequency calculation of the ground states for the neutral and cationic [4]helicene monomer were performed at the PBE0[25]/def2-TZVPP[26] level of theory. Similar calculations for the excited states of the [4]helicene cation were done using the TD-DFT approach at the same level of theory. Based on these energies and harmonic frequencies, vibrationally resolved photoelectron spectra were computed using the ezFCF programme[49].

The conformational space of the [4]helicene clusters was sampled using ABCluster[28,50,51] and Coalescence-Kick software[29,52]. The resulting structures were then optimised at the GFN2-xTB level of theory[30], and a unique set of structures was found using the home-written script, available from the MOLINC repository[53]. Based on these geometries, vertical ionisation potentials (vIPs) were evaluated using various ab initio approaches (DLPNO-CCSD(T)[32–34], IP-EOM-DLPNO-CCSD[35–37], subsystem-based $G_0W_0$@BHLYP[42,54,55], and FDE-ET[38–41] with cc-pVTZ[56] basis set) and semi-empirical methods (DFTB-CI[22,43], and IP-xTB2[57]).

To find the racemization barrier of [4]helicene monomer and dimer, the transition states for racemization were optimised at the PBE0-D3(BJ)/def2-TZVPP and PBE0-D3(BJ)/def2-SV(P) level of theory, and single point energies of the equilibrium structures and transition states were recalculated at the DLPNO-CCSD(T)/cc-pVTZ level of theory (see Supplementary Fig. 1).

All calculations used ORCA[58], SERENITY[59,60], XTB[57], or deMon-Nano[61] quantum-chemical packages. Detailed descriptions of each of the types of calculations are available in the supplementary computational details.

## Data availability

All the structural data is provided in the supplementary information file and in the supplementary data 1. Computational details and analyses that supported this study are available within the main text and the supplementary information, and also available from the corresponding authors on request.

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

## Acknowledgements

This work has been funded by the Deutsche Forschungsgemeinschaft (DFG, German Research Foundation)—Projektnummer 328961117—SFB 1319 ELCH. S.R.D. acknowledges funding from FCT Portugal through grant UIDB/04564/2020 (https://doi.org/10.54499/UIDB/04564/2020) and UIDP/04564/2020 (https://doi.org/10.54499/UIDP/04564/2020). FCT funding via the Scientific Stimulus Employment Programme is also acknowledged (CEECIND/01895/2018/CP1585/CT0002). K.M and N.A. acknowledge the CNRS, the University of Angers, and the Région Pays de

la Loire through the RFI LUMOMAT (grant to K.M.). P.E. and J.N. gratefully acknowledge funding by the Deutsche Forschungsgemeinschaft (DFG, German Research Foundation) through IRTG 2678 Münster-Nagoya (GRK 2678—437785492). H.R.H. acknowledges support by the Programme National "Physique et Chimie du Milieu Interstellaire" (PCMI) of CNRS/INSU with INC/INP co-funded by CEA and CNES. H.R.H. acknowledges also the COST Action CA21126 - Carbon molecular nanostructures in space (NanoSpace), supported by COST (European Cooperation in Science and Technology). P.E. thanks Johannes Tölle for helpful discussions for carrying out the subsystem $G_0W_0$ calculations. The authors acknowledge the use of the GWDG and Maxwell computer clusters. The authors thank the general staff at Synchrotron SOLEIL for smoothly running the facility and for the provision of beamtime under project number 20181109. Co-funded by the European Union (ERC, 101040850-MiCRoARTiS). Views and opinions expressed are however those of the author(s) only and do not necessarily reflect those of the European Union or the European Research Council. Neither the European Union nor the granting authority can be held responsible for them.

## Author contributions

S.R.D., A.L.S. and M.S. conceived the project. S.R.D., A.L.S, S.G., H.R.H. carried out the experiments. D.S.T., P.E and S.R.D. carried out the theoretical calculations. S.R.D., D.S.T., P.E and H.R.H. analysed the data. K.M. synthesised all the samples used. G.A.G., L.N., M.S. and N.A. supervised all the experimental and synthetic work. J.N. supervised all theoretical calculations. All authors contributed to interpretation of the results. S.R.D. and M.S. co-wrote the manuscript with feedback from all authors.

## Funding

## Competing interests

The authors declare no competing interests.
