## [Peer Review File · Nature Communications]

Evolution of the ionisation energy with the stepwise growth of chiral clusters of [4]heliceneReviewer #1 (Remarks to the Author):

This is an interesting manuscript describing a combined experimental and theoretical study of small chiral [4]helicene clusters. The paper is clearly written, and the theoretical and experimental results are well matched. It is argued that, as determined from the experimental ionization onsets and theoretical calculations, the aggregation of homochiral clusters is favored over heterochiral clusters with increasing cluster size. The results are convincing, and that the results will be of interest to a significant number of researchers, and I think it should be published in Nature Communications. I have a number of minor comments and questions that the authors should consider before the paper is accepted.

1. The authors need to include a formula, or better, a picture of the [4]helicene molecule as early on in the paper as possible. A lot of people who might be interested in this paper will not know what [4]helicene looks like.
2. How stable are the chiral helicene species to racemization? Is this rate very slow? Is it possible that in the dimers, or larger clusters, that there is a kind of templating or catalysis that lowers the barrier to this isomerization, so that what might start out as a heterochiral cluster becomes homochiral, or are the barriers just way too high?
3. The authors state that the agreement between the theory and experiment in Figure 2 is "noticeably good." To me it seems okay, but the theory for the first band seems too low by ~ 0.3 eV, and perhaps more intensity would be expected in the highest energy band in the calculation. It might be useful to indicate how good the theory is expected to be.
4. Why did the authors choose a photon energy of 11 eV to record the mass selected photoelectron spectra of Figure 3? I realize beamtime is limited, but a different choice of photon energy might have resulted in a higher ionization probability to produce lower energy states of the cation. (For example, in general, autoionization will populate very different final states from direct photoionization.) Is there any rationale for 11 eV?
5. Is the sample a completely racemic mixture of [4]helicene? If so, it should be stated.

Reviewer #2 (Remarks to the Author):

The manuscript presents new mass-selected photoelectron spectra on the first seven helicene clusters, supplemented by quantum-chemical calculations. The photoelectron photoion coincidence experiments are state of the art and probably nobody could do better, but the way they are analysed, presented and, especially, compared to the calculations is rather strange. The data is probably publishable, although it may not be the clickbait this particular journal is usually looking for. Some of my specific questions and comments are summarized below.

- The presented PES are neither TPES nor SPES, which the research group is knowledgeable in and which are both vastly superior to the single-photon energy technique used herein. Any particular reason for that? (I would guess signal levels) Could the authors comment on the energy resolution and calibration (as it is definitely energy-dependent in the photoelectron VMI)?
- The lack of noise (or features) on the PES in Figure 3 is surprising. Or, do I see correctly that since the S/N for the monomer is so many times higher than for the clusters, the experimental noise is imperceptible for the former? How about the clusters? Figure 3 is so compressed that it is largely useless to examine the individual spectra, but Figure 4 in the supplementary material shows these spectra with much better detail and there the peaks look too pretty to be just noise. Any comment on their assignment? Are they vibrational progressions? (For example, $n=4$ or 5)
- Since these are mass-selected photoelectron spectra, the question of dissociative photoionization naturally arises. With AIEs less than 7 eV and a photon energy of 11 eV, one would expect dissociation of the larger clusters, i.e. the signal in cluster n might come from cluster $n+1$. There is a short side-note on using translational ion energy filtering to exclude DPI signal, but nothing on

whether this actually worked, whether DPI signal was in fact seen on the ion image. What is the estimated timescale of the dissociation of such large systems vs. the experimental flight times?

- All the above comments are relatively minor, but there is one thing that truly doesn't add up: the calculations are said to produce vertical ionization energies, but they are compared against the experimental adiabatic IEs. Why? Especially for the monomer (and maybe also for the dimer) the vertical IE is much easier obtained and better defined experimentally.
- There are few language issues, but one needs mentioning: the authors surely did not mean precision, rather accuracy when they commented on the DLPNO-CCSD(T) method.

Reviewer #3 (Remarks to the Author):

Dear editor and authors,

After carefully reading the manuscript 'Evolution of the ionisation energy with the stepwise growth of chiral clusters of [4]helicene' I come to the conclusion that it reports on correctly executed research and after considering some of the comments below it could be published but probably not in a journal serving a scientific community as broad as that of nature communications.

Comments:

On page 1:

See also main comment above. The authors, in general, fail to convince me of the importance of this work for a broad audience.

On page 4:

'where an increasing abundance for the following larger aggregates is registered.' Is this truly significant?

On page 5:

'Overall, the agreement between the calculated vibronic progressions and the experimental spectrum is noticeably good,' this seems a bit of an optimistic statement. To me the calculated spectrum captures most of the experimentally observed features to a certain level at best.

On page 6, figure 3

The spectrum of the monomer seems to have a much larger intensity. Could the authors comment on this? Also, from the monomer to the dimer we see a strong increase in details but afterwards this level seems decreases again. It is not clear to me what is happening here.

On page 7 caption of fig 4.:

$(n(P) - n(M))/(n(P) + n(M))$ is $n(P)$ always strictly larger than $n(M)$?

On page 8 figs 5 and 6:

Maybe add a comment of what the different shades of grey mean?

On page 9:

I would suggest to add some more details on the computation methods in the main manuscript. Especially on what is the reference system on which the GOW0 is build.

Supplementary:

Please provide the structure data in separate text files. Numerical data in tables in PDF's is a recipe of disaster cost by copy past errors.

Response letter "Nature Communications manuscript NCOMMS-23-43827"

"Evolution of the ionisation energy with the stepwise growth of chiral clusters of [4]helicene"

We thank the editor and referees for the time spent reading and critically evaluating our work, as well as for the opportunity given to submit a revised manuscript. We appreciate the positive remarks, constructive comments, and suggestions from all three reviewers on how to improve our manuscript to reach further the audience of Nature Communications. Below we provide a point-by-point response to the reviewers' comments. All changes in the manuscript are highlighted in red in the revised manuscript and supplementary information files. Below, we list a few notable changes in the revised manuscript and supplementary information files.

1. New figure added (Fig. 1 in page 3 of the revised manuscript).
2. New figure added (Fig. 1 in page 6 of the Supplementary Information).
3. New table added (Table 7 in page 16 of the revised Supplementary Information).
4. The "Results" section (2) has now two subsections (2.1 Photo-electron photo-ion coincidence spectroscopy and 2.2. Theoretical modelling of [4]helicene clusters).
5. Following the "Discussion" section (3), a "Methods" section (4) was added (pages 12-13 of the revised manuscript), containing two subsections (4.1. Experimental details and 4.2. Computational details). Here, a condensed summary of the experimental and theoretical methodology is presented, with reference to the Supplementary Information for further details.
6. A new subsection was included in the Supplementary Information: "Racemization barrier of [4]helicene monomer and dimer" (pages 6-8 of the revised SI).
7. Structural data for the relevant structures given in table 2 of the SI are now provided as "Supplementary Data". A ZIP folder "StructuresBelow210wvnThr" has been prepared, including all the structures of the [4]helicene clusters calculated at the GFN2-xTB level of theory that are below 210 cm⁻¹ in energy for each combination of enantiomers. The file names correspond to the type of the structure, consistent with the naming used in the main article and SI.

Reviewer #1 (Remarks to the Author):

This is an interesting manuscript describing a combined experimental and theoretical study of small chiral [4]helicene clusters. The paper is clearly written, and the theoretical and experimental results are well matched. It is argued that, as determined from the experimental ionization onsets and theoretical calculations, the aggregation of homochiral clusters is favored over heterochiral clusters with increasing cluster size. The results are convincing, and that the results will be of interest to a significant number of researchers, and I think it should be published in Nature Communications. I have a number of minor comments and questions that the authors should consider before the paper is accepted.

1. The authors need to include a formula, or better, a picture of the [4]helicene molecule as early on in the paper as possible. A lot of people who might be interested in this paper will not know what [4]helicene looks like.

We thank the referee for this suggestion. We have prepared a new figure (Fig. 1, page 3 of the revised manuscript), depicting the 3D structure of the *P* and *M* enantiomers of [4]helicene, with a perspective that favours the perception of the intrinsic helical chirality.

2. How stable are the chiral helicene species to racemization? Is this rate very slow? Is it possible that in the dimers, or larger clusters, that there is a kind of templating or catalysis that lowers the barrier to this isomerization, so that what might start out as a heterochiral cluster becomes homochiral, or are the barriers just way too high?

This is an interesting remark by the referee. In a recent theoretical study on the racemization barriers of helicenes (Barroso, J. et al. Revisiting the racemization mechanism of helicenes. Chem. Commun. (Camb.). 54, 188–191 (2018)), the authors calculate a racemization barrier for [4]helicene from the achiral transition state to be about 16.7 kJ/mol. To compare these numbers with our preferred methods we performed a new calculation: the equilibrium structure (ES) and a planar transition state (TS) of the neutral [4]helicene were optimized at the PBE0-D3(BJ)/def2-TZVPP level of theory using ORCA 5 software. For both the optimized structures at DLPNO(TightPNO)-CCSD(T)/cc-pVTZ, single-point energies were computed. The electronic energy difference between the TS and ES was found to be 18 kJ/mol (0.19 eV) at both PBE0-D3(BJ)/def2-TZVPP and DLPNO(TightPNO)-CCSD(T)/cc-pVTZ levels of theory. The zero-point-energy-corrected barrier height is 17.3 kJ/mol (0.18 eV), and the Gibbs free energy barrier at 298 K is 18.3 kJ/mol (0.19 eV).

This energy barrier is easily overcome in our oven (heated to a temperature of about 200°C) by thermalization of all its vibrational modes followed by IVR, and therefore there is a fast equilibrium of *M* and *P* enantiomers. However, during the adiabatic expansion the molecular beam reaches temperatures below 100 K, and the molecular clusters that are produced have only a few tens of microseconds until they are observed. At these low temperatures the enantiomers will be energetically trapped, and the racemization process rate should be considerably lower. To confirm this, we performed an additional calculation where we determined the interconversion barrier for *PP*⇌*PM* dimers. For the two optimized equilibrium states of *PP* and *PM*, a climbing image nudged elastic band method (CI-NEB) TS search and consecutive harmonic frequency calculation were performed (see full details in pages 6-8 of the revised SI). The results reveal an activation barrier for racemization of one [4]helicene monomer in the dimer in the range of 23 to 27 kJ/mol (depending on the level of theory), which is a relevant increase compared with the barrier determined for the monomer (18 kJ/mol). Based on these predictions, a new figure was produced displaying the activation barriers for the monomer and dimer (Fig. 1 of the revised SI). The results strongly suggest that the racemization barrier is very likely increasing with the number of molecules in the cluster since intermolecular interactions will hamper conformational changes. In conclusion, we believe that the [4]helicene clusters observed are indeed stable and no racemization occurs when we perform the spectroscopy measurements. A mention to these points has been added to the Methods section of the revised manuscript and further expanded in the SI.

3. The authors state that the agreement between the theory and experiment in Figure 2 is “noticeably good.” To me it seems okay, but the theory for the first band seems too low by ~0.3 eV, and perhaps more intensity would be expected in the highest energy band in the calculation. It might be useful to indicate how good the theory is expected to be.

Considering the computational cost of the pure DFT approach, and the absence of any energy shifts or scaling factor applied, the 0.3 eV deviation is less than 5% of the experimental value. We consider this to be a reasonably good agreement. For clarity, we rephrased the sentence: see response to the third point raised by reviewer #3.

4. Why did the authors choose a photon energy of 11 eV to record the mass selected photoelectron spectra of Figure 3? I realize beamtime is limited, but a different choice of photon energy might have resulted in a higher ionization probability to produce lower energy states of the cation. (For example, in general, autoionization will populate very different final states from direct photoionization.) Is there any rationale for 11 eV?

We did record a TPES and TIY for the [4]helicene monomer, from threshold up to 9.35 eV. The TPES did not show any vibrational structure, so we decided to record fixed photon energy spectra (PES) instead. Indeed, if one is not looking for resolution, the PES has some advantages over TPES, namely, lower acquisition times (typically ten-fold), insensitivity to source instabilities—an issue when working with very low product quantities such as in this case—and access to anisotropy parameters (out of this article's scope). Time constraints typical of large facilities compelled us to compromise between the VMI resolution and bandwidth, by adjusting the electrostatic extraction field so that 100 % electron are collected, and we found 11 eV to be optimal since it allowed recording the first three photoelectron bands while keeping a high signal-to-background ratio. The latter decreases with increasing photon energy due to processes such as photoemission from metallic surfaces and/or ionization of the background gas, especially when high densities cannot be achieved in the molecular beam.

5. Is the sample a completely racemic mixture of [4]helicene? If so, it should be stated.

This point is clarified in the “Methods” section (4.1 Experimental details in page 12 of the revised manuscript) where we added the following sentence “A sample of racemic [4]helicene was placed in a home-built stainless-steel oven (in-vacuum) and heated to approximately 200 °C to generate sufficient vapour pressure. Here, the molecules have time to accumulate thermal energy in the vibrational modes to overcome the activation barrier of around 18 kJ/mol following intramolecular vibrational redistribution, so that an equilibrium between P and M enantiomers is expected.” For further details see also response to point 2 in the previous page.

Reviewer #2 (Remarks to the Author):

The manuscript presents new mass-selected photoelectron spectra on the first seven helicene clusters, supplemented by quantum-chemical calculations. The photoelectron photoion coincidence experiments are state of the art and probably nobody could do better, but the way they are analysed, presented and, especially, compared to the calculations is rather strange. The data is probably publishable, although it may not be the clickbait this particular journal is usually looking for. Some of my specific questions and comments are summarized below.

- *The presented PES are neither TPES nor SPES, which the research group is knowledgeable in and which are both vastly superior to the single-photon energy technique used herein. Any particular reason for that? (I would guess signal levels) Could the authors comment on the energy resolution and calibration (as it is definitely energy-dependent in the photoelectron VMI)?*

Both referees make a good point (see our answer to R1 above). There are several advantages to TPES over PES, including a better and constant energy resolution, but also a more precise calibration of the energy scale. We decided to work at fixed photon energy because the monomer TPES (recorded with a low backing pressure) show no vibrational structures that would require a high electron energy resolution. Additionally, we had some stability issues when producing clusters linked to the higher temperature needed to produce enough density for clusters to be made (nozzle/skimmer blocking), so that PES was better adapted, being insensitive to source variations, as compared to a long duration photon-energy scan as required for TPES recording.

While with TPES we can typically give an absolute energy scale with a precision of a few meVs, for PES it is of the order of 10 meV for the lower binding energies, due to its quadratic relation between the electron kinetic energy and the radius of the VMI's patterns. As for the resolution, it is indeed kinetic energy dependent, and it needs to be convolved with the 28 meV photon energy resolution, given by the quite open slits of the monochromator. In the 7 eV binding energy region and for a photon energy of 11 eV, the electron energy resolution is estimated at 110 meV from the data in Ref. 25 leading to total energy resolution of 114 meV. The absolute resolution improves for slower photoelectrons, approaching 20% of eKE for the high binding energies according to the published curve.

• *The lack of noise (or features) on the PES in Figure 3 is surprising. Or, do I see correctly that since the S/N for the monomer is so many times higher than for the clusters, the experimental noise is imperceptible for the former? How about the clusters? Figure 3 is so compressed that it is largely useless to examine the individual spectra, but Figure 4 in the supplementary material shows these spectra with much better detail and there the peaks look too pretty to be just noise. Any comment on their assignment? Are they vibrational progressions? (For example, n=4 or 5)*

Because the spectra are obtained by Abel transformation of the photoelectron image, and because the Abel is an inverse function, noisy images might lead to artefacts in the final PES (for instance oscillations as a function of eKE). It is not always trivial to separate artefacts from real signal in very noisy images. Although assigning individual spectral features requires electronic structure calculations which are out of the scope of this article for cluster species, we have nevertheless tried to answer the referee's comment by changing the signal-to-noise ratio in the pixels and looking at statistical error bars. Indeed, one way of determining real structure is to change the image statistics by binning several pixels together. This decreases the experimental resolution but increases the pixels statistics and therefore decreases the oscillatory behavior. Error bars are calculated by assuming a Poisson distribution in each pixel (*i.e.*, variance = value), since we are counting particle hits at a given position/pixel, and then applying error propagation formulae throughout the algebra operations within the Abel algorithm. Real features will persist when changing the value of the binning. Looking at the resulting spectra (see figure below) and with the addition of the error bars, some features seem to be real. Whether they correspond to vibronic structure, or to different conformer families or even to closely spaced electronic states, it is difficult to establish without properly simulating each individual PES, which is not realistic for these systems. Note that, as the referee points out in the next comment, assigning the whole spectra might be further complicated by the presence of DPI.

Considering this discussion, we thought it was pertinent to add the following sentence in the main text: *"Note that some of the oscillations visible in the clusters' spectra might be due to noise induced by the Abel inversion of the VMI images, but also to possible vibrational progressions, to the presence of various conformation families or even to closely lying electronic states whose analysis is out of the scope of the present work."* (page 7, lines 10-14 of the revised manuscript).

- *Since these are mass-selected photoelectron spectra, the question of dissociative photoionization naturally arises. With AIEs less than 7 eV and a photon energy of 11 eV, one would expect dissociation of the larger clusters, i.e. the signal in cluster n might come from cluster n+1. There is a short side-note on using translational ion energy filtering to exclude DPI signal, but nothing on whether this actually worked, whether DPI signal was in fact seen on the ion image. What is the estimated timescale of the dissociation of such large systems vs. the experimental flight times?*

Measuring DPI on H-bonded or even non-H bonded clusters is doable (see <https://dx.doi.org/10.1039/d2cp05679h>) but challenging. Even if there are differences in translational energies, one cannot distinguish direct and dissociative ionization at just one photon energy because the neutrals might also be produced through evaporation events in the molecular beam, before they are ionized. A photon energy dependent study needs to be performed which is clearly out of the scope of the present study. Indeed, in this article, we are only interested in the ionization energy onset, where we can assume that no DPI takes place. In another π -stack system such as pyrene, dissociative energies (difference between IE and fragment apparition energy) of ≥ 1 eV have been observed. From the energies in tables 2 and 6 of the SI, one can extract the calculated dissociation energies for the process $M_n^+ \rightarrow M_{n-1}^+ + M$ for the most stable cluster at 1.04 eV for n=2 and 0.81 eV for n=7 for instance. As the referee points out, these values are most likely lower limits due to the very short analysis time of the spectrometer (few μ s), which translates into a so-called kinetic shift of the DPI process towards higher ionization energies. Therefore, we believe that our onset values are not affected by DPI, since each cluster should be stable at least over the few 100's meV above its IE. We have, however, changed the manuscript accordingly to reflect this discussion (see below), and we have added a Table in the supplementary information (Table 7, page 16 of the revised SI file) with the calculated dissociation energies for the neutral and cationic clusters.

The sidenote in page 4 of the original manuscript has been removed. In the discussion of the results, we now introduced a new paragraph (page 7, lines 1-10 of the revised manuscript): *“Although mass selection does not necessarily translate to size selection due to dissociative photoionisation processes (DPI), we can safely assume that this will be the case close to the ionization onsets. Indeed, in another π -stack system, namely pyrene, dissociative energies (difference between IE and fragment appearance energy) of more than 1 eV have been observed. Here, from the energies in tables 2 and 6 of the SI, one can extract the calculated dissociation energies for the process $M_n^+ \rightarrow M_{n-1}^+ + M$ for the most stable cluster at 1.04 eV for n=2 and 0.81 eV for n=7 for instance (see Table 7 of the SI). These values are most likely lower limits due to the very short analysis time of the spectrometer (few μ s), which translates into a so-called kinetic shift of the DPI process towards higher ionization energies and to the possible existence of reverse barriers. Therefore, we believe that our onset values are not affected by DPI, since each cluster should be stable at least over the few 100 meV's above its IE.”*

- *All the above comments are relatively minor, but there is one thing that truly doesn't add up: the calculations are said to produce vertical ionization energies, but they are compared against the experimental adiabatic IEs. Why? Especially for the monomer (and maybe also for the dimer) the vertical IE is much easier obtained and better defined experimentally.*

The referee is correct, but comparison is challenging and therefore we have chosen the lesser of few evils. On the experimental side, we can only see a clear vertical transition in the monomer. For clusters, such a vertical energy determination is not trivial due to spectroscopical congestion and the presence of multiple conformers (see for instance Fig. 6 of the revised manuscript). Therefore, to apply the same methodology across all sizes, we have chosen to extract the observed ionization onset by careful extrapolation of the onset. Note that this is not the AIE because temperature and resolution effects for instance might shift this value to lower ionization energies but, most importantly, it allows us to apply the same methodology to all clusters. This value, however, should be closer to the AIE than to the VIE.

On the theoretical part, calculating AIEs for these systems taking into account the number of conformers studied is too costly because the cation's geometry would also need to be optimized. Thus, only vertical values have been calculated for the clusters, although for the monomer both AIE and VIE are given.

Due to the accuracy of the calculations for these large systems, and the experimental limitations mentioned above, comparison of absolute values is pointless and only the trend with cluster size gives meaningful information. This is true as long as the VIE - AIE difference remains constant with size, as well as the experimental IE_{obs} - AIE difference.

In page 9 of the original manuscript (now page 10, lines 7-12 of the revised manuscript), the following phrase summarizes this response:

“We note that a shift between experimental and theoretical values is expected due to temperature effects (hot bands, presence of isomers) and potential differences between vertical and adiabatic ionisation energies. However, we find that regardless of the intrinsic limitations when comparing ionisation thresholds – obtained from the rising onset – to calculated vertical ionisation energies, the observed ionisation thresholds yield a trend that is consistently captured in the calculated vertical ionisation energies.”

We also note that a similar and meaningful approach was followed for pyrene clusters (see <https://dx.doi.org/10.1039/d2cp05679h>)

- *There are few language issues, but one needs mentioning: the authors surely did not mean precision, rather accuracy when they commented on the DLPNO-CCSD(T) method.*

This issue has now been fixed in the revised manuscript (page 11, line 2).

Reviewer #3 (Remarks to the Author):

Dear editor and authors,

After carefully reading the manuscript ‘Evolution of the ionisation energy with the stepwise growth of chiral clusters of [4]helicene’ I come to the conclusion that it reports on correctly executed research and after considering some of the comments below it could be published but probably not in a journal serving a scientific community as broad as that of nature communications.

We thank the referee for the remark. We address each of the points raised below.

Comments:

On page 1:

See also main comment above. The authors, in general, fail to convince me of the importance of this work for a broad audience.

As interstellar dust is an important component of the ISM of galaxies, dust regulates the radiative transfer of photons with energies below 13.6 eV and dominates the spectral energy distribution of galaxies as well as the heating of the gas in diffuse clouds and PDRs (Tielens, *Front. Astron. Space Sci.* (2022) DOI: 10.3389/fspas.2022.908217 and Berné et al. *Astronomy & Astrophysics* 667, A159, (2022) <https://doi.org/10.1051/0004-6361/202243171>). PAH clusters are considered important precursors of carbonaceous dust particles in the ISM (Salama, *Proceedings of the International Astronomical Union* (2008), <https://doi.org/10.1017/S1743921308021960>; Jäger et al. *The Astrophysical Journal* (2009),

DOI: 10.1088/0004-637X/696/1/706) but as their formation mechanism and interactions with VUV radiation are not well understood, it is of great importance for astrochemical models to reveal the dynamics and structure of PAH clusters in the gas phase (Potapov, *Molecular Astrophysics* (2017), DOI: 10.1016/j.molap.2017.01.001). PAH clusters have also been tentatively observed in massive dust-producing Population I Wolf-Rayet binaries which show a wealth of absorption and emission details in the circumstellar dust envelopes (Marchenko & Moffat, *Monthly Notices of the Royal Astronomical Society* (2017) doi:10.1093/mnras/stx563).

We thought pertinent to bring some of these highlights into the introduction section to facilitate a connection with the broad audience of the journal. The last paragraph of the introduction (page 4, lines 17 to 24 of the revised manuscript) has been extended as the following:

“While it is established that PAH clusters are important precursors of carbonaceous dust particles in the ISM, their formation mechanisms and interactions with VUV radiation are not well understood. Thus, it is of great importance for astrochemical models to elucidate the structure and dynamics of PAH clusters in the gas phase. Here, results of a combined experimental and theoretical study on the ionisation energies of chiral clusters of [4]helicene up to the heptamer are presented. Through the introduction of handedness in cluster formation, this investigation extends our understanding of symmetry breaking and chiral amplification of large systems with rich PAH content in harsh vacuum environments.”

On page 4:

‘where an increasing abundance for the following larger aggregates is registered.’ Is this truly significant?

Its significance may be questioned, but we believe this to be certainly noteworthy, therefore we decided to mention it in the main text.

On page 5:

‘Overall, the agreement between the calculated vibronic progressions and the experimental spectrum is noticeably good,’ this seems a bit of an optimistic statement. To me the calculated spectrum captures most of the experimentally observed features to a certain level at best.

The phrasing has been edited to the following: *“Overall, the calculated spectrum captures the essential features observed in the experiment, with the energies of each of the electronically excited states being well predicted and their relative spacing captured.”* (page 6, lines 3-5 of the revised manuscript).

On page 6, figure 3

The spectrum of the monomer seems to have a much larger intensity. Could the authors comment on this? Also, from the monomer to the dimer we see a strong increase in detail but afterwards this level seems decreases again. It is not clear to me what is happening here.

The units in that Figure are arbitrary and not comparable from one mass to another. However, it so happens that the monomer is much more intense, this is normal because in the molecular beam, the probability of forming clusters decreases with size. As for the increase in detail, this is related to the fact that the spectra are obtained from the Abel transform of the photoelectron images. The Abel transform is an inverse function, and as such will amplify the noise of the image, leading to the oscillations observed in the spectra. See answer to a similar comment by R2 above.

On page 7 caption of fig 4.:

$(n(P) - n(M))/(n(P) + n(M))$ is $n(P)$ always strictly larger than $n(M)$?

The quantities $n(P)$ and $n(M)$ can vary from 0 to $n(P)+n(M)$. However, since the photoelectron spectra given here are not sensitive to the chirality of the system, the results for the paired enantiomers with $n(M)>n(P)$ would be the same as for their chiral counterpart. For example, the PPM trimer produces the same PES spectrum as the MMP trimer. Thus, we provide only one set of values (where $n(P)>n(M)$), since their mirror images produce the same results.

To clarify this point, we added the following sentence to the main article: *“Here, only the results for the paired enantiomers with $n(P)>n(M)$ are shown, as they would be the same as for their chiral counterpart with $n(M)>n(P)$. For example, the PPM trimer produces the same PES spectrum as the MMP trimer.”* (page 8, lines 4-6 of the revised manuscript).

On page 8 figs 5 and 6:

Maybe add a comment of what the different shades of grey mean?

We added the note *“Different shades of grey are used to facilitate visualisation of monomeric elements within the larger clusters.”* in the caption of Figures 6 and 7 of the revised manuscript to remove any ambiguity. The decision to add different shades of grey was solely to serve as an aide to the eye, facilitating the visualization the monomeric elements within the larger clusters. The correct enantiomeric form is always labelled as either P (grey) or M (red).

On page 9:

I would suggest to add some more details on the computation methods in the main manuscript. Especially on what is the reference system on which the GOWO is build.

The referee is right. In the original submission, we considered that a full description of the theoretical treatment would overwhelm the flow of the manuscript and decided to detail the computational methods in the supplementary information. We have now prepared a condensed summary of the experimental and computational methods and included it in the “Methods” section, now included at the end of the revised manuscript. Additionally, we thought it was pertinent to add a small edit in the revised manuscript, where for clarity, we changed the mentions of GOWO to "GOWO@BHLYP" (page 10, line 15 of the revised manuscript).

Supplementary:

Please provide the structure data in separate text files. Numerical data in tables in PDF's is a recipe of disaster cost by copy past errors.

We appreciate this remark. The structures and numerical experimental data are now provided in separate text files. A ZIP folder has been prepared and submitted as an attachment with the revised manuscript.

Reviewer #1 (Remarks to the Author):

The authors have addressed my (Referee 1) comments and questions. I was also asked to comment specifically on the response to referee 2's comments.

From my perspective, the authors have addressed Referee 2's comments adequately, and I believe the paper should be accepted. Here are some specifics.

Referee 2 Comments:

Comment 1 (summarized) and response: Why PES and not TPES/SPES?

While TPES/SPES have advantages, as the authors point out, they require a stable source over the duration of the wavelength scans. For these experiments, the authors indicate the source is not very stable, and the choice of techniques seems to me to be the correct one. The resolution of the PES is sufficient to address the issues of the paper.

Comment 2 and response: Noise or lack of noise in the PES.

The authors address this question adequately as well. I think the perceived noise (or lack thereof) is sometimes difficult to assess when the data is transformed or processed – in this case the Abel transform moves the noise around. The authors have addressed this in the revised text, and I think they have a good response.

Comment 3 and response: regarding dissociative ionization.

The authors acknowledge that this is a potential issue, but because they are interested in the region around the ionization onset, they argue that dissociative ionization is not a problem for their interpretation. This conclusion is supported by comparison with a similar system (pyrene), and the argument seems reasonable to me. The authors have added text to clarify this in the manuscript.

Comment 4 and response: Theoretical vertical IE vs. Experimental ionization onset

The referee pointed out the theory provided vertical IEs and the experiment provided ionization onsets, and wondered why one didn't compare theoretical vertical IEs to experimental vertical IEs. This is a reasonable concern. The authors indicate that measuring the vertical IE is difficult for a number of reasons (to which I would add that the vertical IE could be above the dissociative ionization threshold), and that calculating the adiabatic IE is considerably more difficult than calculating the vertical IE. They argue that the trends in the experiment (onset) and theory (VIE) show similar behavior. They have added text to clarify this comparison, and I think their arguments are convincing enough. My only worry is that since they are not measuring the adiabatic ionization thresholds, and the observed ionization onsets are higher than this, one might start to wonder again about dissociative ionization – i.e., how far are the observed onsets from the dissociative ionization thresholds? Personally, I don't think this is a problem, but it might be worth mentioning.

Overall, while I cannot really speak for Referee 2, I feel the authors have adequately addressed Referee 2's comments, and that the manuscript should be accepted for publication.

Reviewer #3 (Remarks to the Author):

Dear author and editors,

The reply and revised manuscript respond to all my previous comments in a sufficient manner. I still think the introduction and abstract could be improved to appeal to a broader public but I assume this is beyond providing feedback on the soundness of the research reported. My recommendation is to accept this manuscript for publication.